# The Utilization of Cell-Penetrating Peptides in the Intracellular Delivery of Viral Nanoparticles

**DOI:** 10.3390/ma12172671

**Published:** 2019-08-22

**Authors:** Jana Váňová, Alžběta Hejtmánková, Marie Hubálek Kalbáčová, Hana Španielová

**Affiliations:** 1Department of Genetics and Microbiology, Faculty of Science, Charles University, Viničná 5, 128 44 Prague 2, Czech Republic; 2Institute of Pathological Physiology, 1st Faculty of Medicine, Charles University, U Nemocnice 5, 128 53 Prague 2, Czech Republic

**Keywords:** cell-penetrating peptide, protein transduction domain, viral particle, intracellular delivery

## Abstract

Viral particles (VPs) have evolved so as to efficiently enter target cells and to deliver their genetic material. The current state of knowledge allows us to use VPs in the field of biomedicine as nanoparticles that are safe, easy to manipulate, inherently biocompatible, biodegradable, and capable of transporting various cargoes into specific cells. Despite the fact that these virus-based nanoparticles constitute the most common vectors used in clinical practice, the need remains for further improvement in this area. The aim of this review is to discuss the potential for enhancing the efficiency and versatility of VPs via their functionalization with cell-penetrating peptides (CPPs), short peptides that are able to translocate across cellular membranes and to transport various substances with them. The review provides and describes various examples of and means of exploitation of CPPs in order to enhance the delivery of VPs into permissive cells and/or to allow them to enter a broad range of cell types. Moreover, it is possible that CPPs are capable of changing the immunogenic properties of VPs, which could lead to an improvement in their clinical application. The review also discusses strategies aimed at the modification of VPs by CPPs so as to create a useful cargo delivery tool.

## 1. Introduction

Currently, viruses that are commonly considered to be harmful to humans are being investigated as potentially promising therapeutic tools; indeed, a large number of unique characteristics predetermine viruses as multifunctional vectors for the transportation of various substances to cells. During the course of their evolution, viruses have mastered the ability to enter cells efficiently and to overcome all the barriers present both in the cell and the host. Viral particles (VPs) that are devoid of a viral genome are known as virus-like particles and they are harmless and safe. Moreover, VPs encompass an interesting class of biomaterials that have huge potential in terms of their application in the field of nanomedicine. VPs are accessible in a wide variety of geometries that exhibit several important features: (i) they are formed via a self-assembly process from structural subunits, and, thus, usually have a uniform size (in the range from 20 nm to 200 nm, depending on the virus species); (ii) they have the potential for large-scale production; (iii) they evolved as nanocontainers for the efficient transport of nucleic acids (DNA, RNA) in biological fluids under the surveillance of the host immune system; (iv) they exhibit efficient interaction with target cells and utilize various endocytic pathways for the intracellular delivery of their payloads; (v) their metastability enables the protection of the cargo in the external environment and its intracellular release in a well-defined manner, thus ensuring that the cargo (DNA or RNA) penetrates into the appropriate site of action. Moreover, naturally occurring VPs can be modified by means of genetic engineering techniques or functionalized with peptides, fluorescent dyes, polymers, carbohydrates, and oligonucleotides by bio-conjugation reactions. All these characteristics predestine VPs to succeed in the clinical setting and render them an important experimental tool that may assist in the identification of barriers to the intracellular delivery of non-viral nanomaterials and provide direction in terms of design aimed at enhancing their efficiency. Although particles of non-viral origin (e.g., polymer particles and liposomes) are currently being developed and optimized so as to attain the efficiency of viral vectors, these remain still the most frequently used delivery vehicles for clinical applications. Up to 2018, around 70% of all vectors used in gene therapy clinical trials were based on viruses, principally adenoviruses (Ad) and retroviruses [1].

VPs have been predominantly exploited in three ways. Firstly, following the removal of their genetic information, they can be used as containers that can be loaded with a diagnostic substance or a drug (e.g., a small molecule, protein, DNA, or RNA, as reviewed in [2]). Secondly, their surfaces can be modified, e.g., with epitopes for immune recognition in order to develop vaccines (as reviewed in [3,4,5]). Finally, natural or recombinant viruses with viral genetic information hidden inside the capsid can be used as oncolytic viruses for the destruction of target (tumor) cells via replication within them [6].

Nevertheless, a number of challenges remain in terms of enhancing the therapeutic potential of VPs. In order to increase the capacity and to ensure the maximal safety of viral vectors, a lot of viral features need to be removed. Consequently, the attainment of the optimal biosafety profile usually comes at the expense of efficiency. Moreover, each virus usually enters only a limited range of cell types as determined by the presence of specific receptors (receptor-dependent tropism) that are used by particular viruses for their interaction with specific cells. In addition, a very high degree of cellular entry efficiency is sometimes required in clinical practice so as to ensure that all the targeted cells are affected. Attempts to preserve the safety and to improve the applicability of viral vectors have involved the introduction of a number of innovative modifications. The various attributes of cell-penetrating peptides (CPPs) can be utilized to supply or restore the ability to penetrate cells, broaden the range of target cells, or increase transduction efficiency. This paper presents a review of several key studies concerning the use of CPPs in combination with viral particles aimed at enhancing the versatility of VPs in therapy.

## 2. Cell-penetrating Peptides

CPPs, also known as protein transduction domains (PTDs), are short peptides (usually <30 amino acids (AA) long) with a composition that enables them to penetrate through cellular membranes and that is capable of assisting in the translocation of various substances into the cell. The first to be discovered and the most extensively studied CPP originates in Tat protein, a transactivator of the transcription of human immunodeficiency virus 1 (HIV-1). The Tat peptide consists of a 12 AA-long section of the Tat protein that is responsible for the translocation of the whole protein across the plasma membrane and into the cell [7,8]. Since the discovery of the Tat peptide in 1988, hundreds of novel CPPs derived from both viral and non-viral proteins have been examined (as reviewed in [9]). In 2016, the repository of experimentally validated cell-penetrating peptides contained 1850 peptide entries of various origin [10]. A comprehensive and detailed summary of CPP research as well as the methodology for investigation of CPPs can be found in [11]. Table 1 presents an overview of CPPs discussed in the text.

### 2.1. Classification of CPPs

CPPs are often divided into two main groups according to their AA composition, i.e., cationic and amphipathic (Figure 1); in addition, a class of hydrophobic CPPs is sometimes mentioned [12].

Cationic CPPs generally contain a large number of positively charged AA residues (arginine, lysine, and histidine, Figure 1a). Arginine is considered to be more favorable than lysine due to the interaction of its guanidium group with the anionic components of the plasma membrane [13,14]. Histidine-rich peptides constitute a special class of CPPs; unlike arginine and lysine, histidine is not charged at neutral pH and only in the case of its entry into acidified cellular vesicles (endosomes) does it become protonated and able to compromise the endosomal membrane. This may be caused either by the penetration of the membrane by the histidine-rich peptide (acting as a typical CPP) or via the “proton sponge” effect mediated by the attraction of chloride ions followed by water into the vesicle, thus resulting in vesicle rupture [15].

Amphipathic CPPs constitute a hydrophilic and a hydrophobic part (Figure 1b). The hydrophilic part contains polar AA residues, mainly arginine and lysine, and serves for interaction with the negatively charged phosphate groups, while the hydrophobic part consists of nonpolar AA residues such as leucine, isoleucine, alanine, and valine and participates in the interaction with the hydrophobic part of the phospholipid membrane. The hydrophilic and hydrophobic parts may either be distributed directly next to each other in a linear sequence (primary amphipathic CPPs); or concentrated into separate areas of the secondary structure of an α-helix or also, in a minority of cases, of a β-sheet, although they are dispersed throughout the primary sequence (secondary amphipathic CPPs).

Various other classification schemes have been applied to CPPs. However, a number of CPPs fall into several different classes; thus, in general, they cannot be strictly classified. For example, the LAH4 peptide is described as a cationic amphipathic histidine-rich peptide [16], thus rendering strict categorization impossible. For the purposes of this review, we restricted the classification to the consideration of the primary characteristic and presumed mode of action, i.e., cationic, amphipathic, osmolytic (histidine-rich), and targeting.

### 2.2. Internalization of CPPs

The mechanism via which CPPs penetrate cellular membranes remains the subject of debate [17,18,19]. Initial experiments with cationic Tat and secondary amphipathic penetratin (Pen) revealed that the internalization process can occur at 4 °C, is not dependent on energy, and leads to the cytoplasmic and nuclear distribution of CPPs [20,21]. Therefore, it was believed that CPPs crossed the plasma membrane via direct penetration, concerning which a variety of mechanisms was proposed, including the formation of a toroidal or barrel pore, a carpet-like model, membrane thinning, and the formation of inverted micelles (as reviewed in [17]). Subsequently, however, a number of methodological limitations to CPP research have come to light. Several research groups have demonstrated that the fixation of cells dramatically changes the distribution of CPPs in the cell as compared to live cell experiments where a punctate CPP pattern (suggesting localization in endocytic vesicles) was observed [22,23,24,25]. Studies on living cells have further determined that the internalization of CPPs is an energy-dependent process that occurs via various endocytic pathways: macropinocytosis, clathrin-mediated endocytosis, lipid-raft mediated endocytosis, and others [26,27]. It is supposed that the association of CPPs with the plasma membrane is based on the interaction of the positively charged parts of CPPs with the negatively charged phosphate groups of phospholipids and the other negatively charged components of the plasma membrane, especially glycosaminoglycans such as heparan sulfate proteoglycans. Furthermore, it has been suggested that the attachment of CPPs that are rich in arginine residues may lead to the establishment of a local membrane curvature and endocytosis [14,28].

Nevertheless, the direct penetration mechanism has not been totally excluded since it has been found that it may occur under specific circumstances, e.g., a high dose of CPP [26,29], the CPP type, and/or its cargo [30,31]. It has been suggested that CPPs are able to locally increase the membrane potential which may, in turn, lead to the opening of membrane channels through which CPPs are able to penetrate into the cell [32]. However, in principle, it is hard to imagine direct penetration mechanisms with regard to CPPs that carry cargoes as large as VPs.

Moreover, it is important to mention here that the translocation mechanism may differ depending on the CPP family [17]; the presence of a cargo is able to completely change the internalization pathway of a particular CPP [33,34,35], and, in addition, the entry mechanism and efficiency of entry may vary between different cell lines [36,37,38,39]. In this respect, the combination of CPPs with virus-derived cargoes with predetermined trafficking may help to elucidate the key factors that influence alternation in the internalization pathways of CPP-based delivery systems.

It should be pointed out that conducting a flawless experiment aimed at the exploration of the CPP-cargo entry mechanism is not a simple task. Particular care is required concerning, for example, the quantification of internalized fluorescently-labelled CPPs by means of flow cytometry, which has been shown to require the treatment of cells with trypsin so as to remove the surface-bound CPPs. The absence of trypsin treatment or the fluorescence-quenching of surface bound fluorophores leads to the flow cytometry providing a flawed estimate of CPP uptake because CPPs are usually firmly bound to the cell membrane from the outside [25]. In addition, fluorescent labels have been found to change the internalization pathway and overall properties of CPPs (e.g., their capacity to translocate a cargo) [40,41].

Thus, each system has its specific features and it is difficult to draw any general conclusions about the entry mechanism of CPPs with or without cargoes.

## 3. The Association of VPs and CPPs

It is possible to use the ability of CPPs to penetrate cellular membranes to ensure or improve the translocation of various substances into cells (as reviewed in [42]), including VPs. Complexes of CPPs and VPs do not necessarily follow the internalization pathway typical for the particular VP type, but may exploit a different route (often macropinocytosis) which is more convenient in terms of enabling the transportation of these complexes to the cytoplasm and/or nucleus. Several CPP-VP association approaches can be employed for the preparation of such complexes (Figure 2). Since each approach has its advantages and disadvantages, it may be possible to tailor a strategy that best suits a particular application.

### 3.1. Noncovalent—Electrostatic Interactions between VPs and CPPs

The easiest method via which to create a VP-CPP complex is to make use of the charges of the CPPs (usually cationic) and the VPs (often anionic); their charges differ enough so as to allow for the formation of a complex via electrostatic interactions induced by means of the simple coincubation of VPs with CPPs in a solution (Figure 2a). The cell treatment procedure with VP-CPP complexes varies from study to study. In some cases CPPs and VPs have been mixed together and applied to the cells immediately [43], whereas several studies have involved the initial coincubation of the CPPs and VPs (usually for 5–30 min at RT or 37 °C or even at 4 °C) so as to form a complex followed by cell application [43,44,45,46,47,48,49]. Another study has involved the initial treatment of the cells with CPPs followed by the addition of the VPs [50]. However, the latter approach is probably more suitable for in vitro or ex vivo procedures than for in vivo administration. A notable study by Gratton et al. [46] involved the testing of whether any differences were apparent following the addition of CPPs to cells before, after, or at the same time as the cell application of VPs. They found that while the addition of the Pen peptide before that of an adenovirus led to the enhancement of transduction, the degree of enhancement was markedly less than that following the coincubation of Pen and Ad. The addition of Pen following the application of an adenovirus did not enhance transduction to any degree. Moreover, the study revealed the significant advantage of this CPP-virus complex system in terms of allowing for a decrease in the dose of Ad (5- to 10-fold) in the presence of the Pen peptide so as to provide for the same level of transduction as that provided by Ad alone. This advantage was also confirmed by another study that proved the need for a much lower dose of VPs so as to ensure similar cell transduction rates as those provided in the absence of CPPs [44,46,47,48,51].

The noncovalent association strategy has been successfully employed for various VPs [43,44,45,46,47,48,49,50,51,52] (Table 2). Moreover, it has been shown to be suitable for ex vivo transduction [44,46,48]. The oligomerization of CPPs has been reported to significantly improve several of the characteristics of CPP-VP complexes. Park et al. [48] have demonstrated that the noncovalent association of a branched oligomer of 2–8 CPP (shown for Tat, Hph-1, Pen, and HP4) monomers and recombinant Ad led to a dramatic increase in transduction efficiency even when using a lower dose of Ad compared to monomeric CPP. Up to a 5000-fold lower concentration of tetrameric Tat was sufficient compared to that of the monomeric Tat peptide, thus preventing its potential cytotoxic effect. Moreover, the branched Tat associated with the Ad to a greater extent than its monomeric form did.

However, the noncovalent association of CPPs with VPs approach has a number of limitations. Firstly, due to various unknown variables, it may be difficult to replicate the preparation of such complexes in a uniform manner. Secondly, this approach tends to consume a high amount of the pure CPP, the preparation of which may be expensive, time-consuming, and demanding, depending on the specific CPP. Thirdly, the effect of the biological milieu on the noncovalent interactions between VPs and CPPs is hard to predict. A study by Liu et al. [47] involved the intramuscular injection of the adeno-associated virus (AAV) in a complex with Pen, Tat-HA2, and LAH4 into mice, and revealed that CPPs, in a similar way to cell cultures, promoted the transduction of muscle tissue, thus suggesting that such complexes were stable. However, in other cases, such noncovalent interactions may not prove strong enough to persist within the organism and to resist the dissociation of the VPs and CPPs. The equilibrium between the association and dissociation of noncovalently formed complexes can be crucial; this issue has been described elsewhere [53].

A different approach based on noncovalent association has been applied by Kühnel et al. [50], who took advantage of the high degree of affinity between Ad and its receptor (coxsackie and adenovirus receptor, CAR). They genetically fused Tat or VP22 peptide (a cationic peptide derived from herpes simplex virus 1 [54]) to a soluble part of the CAR and subsequently noncovalently associated the resulting CAR-CPP to an Ad which led to its efficient entry into a broad range of cancer cells independent of the CAR. They confirmed the strong interaction of all the components via its withstanding ultracentrifugation in a density gradient. This approach is yet to be confirmed in vivo.

### 3.2. Covalent—Genetic Modification of VPs with CPPs

Another frequently employed approach to the formation of VP-CPP complexes involves the genetic incorporation of the CPP sequence into that of the viral capsid or envelope proteins (Figure 2b). The production of genetically modified VPs with CPPs is capable of decreasing batch-to-batch variability and increasing CPP stability within the organism and preventing the degradation thereof.

Conversely, the genetic incorporation of CPPs into viral proteins may negatively affect the secondary structure, flexibility, and proper folding of the CPPs that is vital for the preservation of their membrane penetration function. Moreover, the construction and preparation of such vectors is more time-consuming than that of simple coincubation.

The genetic modification of viral proteins with the Tat peptide has been described for VPs derived from Ad [55,56,57,58], baculovirus [59] and the PP7 [60,61], MS2 [62], and lambda [63] bacteriophages (Table 3).

### 3.3. Covalent—Chemical Conjugation of CPPs to the Surface of VPs

VPs are frequently functionalized with CPPs by the means of a covalent linkage (Figure 2b) via reactive groups of lysine, cysteine, aspartate, or glutamate or via “click chemistry” involving the incorporation of unnatural amino acid residues into a protein sequence [64,65,66,67,68,69,70,71,72,73,74] (Table 4). Although it has been suggested that the noncovalent attachment of CPPs to cargo molecules is preferable for the intracellular delivery of a cargo [27,75], Ad with covalently conjugated Tat has exhibited a 1.5 log higher transduction rate than Ad with a noncovalently associated Tat peptide in the same ratio [65]. Ensuring comparable levels of transduction employing the covalent and noncovalent approaches involves the use of large amounts of noncovalently attached Tat [46]. A study by Wang et al. [62] has mentioned that the chemical conjugation of the Tat peptide to the MS2 bacteriophage is more difficult to characterize, more expensive, and less stable than genetic incorporation.

## 4. Modalities of CPP-functionalized VPs

Modification of VPs via CPPs might open up new possibilities in terms of the various applications of VPs, including those applied in a clinical setting. In general, CPPs may help (i) to broaden the spectrum of cell types that internalize VPs with a cargo (e.g., with concern to gene therapy); (ii) to increase the efficiency of the intracellular delivery of VPs with a therapeutic payload; (iii) to modulate the interaction of VPs with immune cells (e.g., regarding vaccine development), and (iv) to efficiently target specific (e.g., cancer) cells (Figure 3). The following text provides a review and discussion of several examples of the utilization of CPPs in combination with VPs.

### 4.1. Broadening the Spectrum of Cells for VP Entry

Various VPs have been investigated with regard to the intracellular delivery of therapeutic payloads, including VPs derived from nonmammalian viruses. Although viral systems derived from plant and insect viruses or bacteriophages are well adapted for interaction with their natural hosts, they do not usually interact efficiently with mammalian cells. That said, a number of exceptions have been identified, e.g., VPs derived from the plant virus cowpea mosaic virus (CPMV) have been found to enter mammalian cells via interaction with vimentin [76] and a prototype baculovirus *Autographa californica* multiple nucleopolyhedrovirus (AcMNPV) has been found to be able to transduce different mammalian cell types (as reviewed in [77]). These viral systems may benefit from modification with CPPs that are able to facilitate the binding and/or internalization of these VPs.

This phenomenon has been demonstrated recently for VPs derived from turnip yellow mosaic virus (a plant-attacking virus) [74] covalently conjugated with Tat: while no native VPs entered the mammalian cells, Tat-assisted transfection was found to be more efficient than lipofection. The modification of VPs derived from the PP7 bacteriophage with CPPs has also been experimentally addressed; unlike unmodified VPs, VPs derived from PP7 with protamine CPP incorporated in a coat protein have been found to be able to enter mouse prostate cancer cells (RM-1), deliver mRNA for a GFP protein, and ensure its expression [61]. Another study has concluded that PP7 VPs modified with Tat peptide successfully packed microRNA precursors (pre-miRNA) which were then protected from nuclease digestion. Following the treatment of SK-HEP-1 hepatoma cells, fluorescently labelled Tat-VPs were detected inside the hepatoma cells, whereas no fluorescence signal was visible with respect to the unmodified VPs. A large increase in the amount of miRNA was evident following the transduction of the cells by the Tat-VPs that carried pre-miRNA, which were then able to decrease the expression of a reporter gene and reduce the migration of hepatoma cells while probably suppressing the liver-intestine cadherin mRNA [60]. Similarly, the Tat modification of VPs derived from the MS2 bacteriophage has been employed for the intracellular delivery of therapeutic RNA [62,70,71]. MS2 VPs genetically modified with the Tat peptide have been shown to efficiently enter tumor cells in mice following their injection into a tail vein and successfully deliver pre-microRNA-122, which led to the inhibition of hepatocellular carcinoma tumor growth in the mice [62]. When MS2 VPs have been modified with the Tat peptide by means of chemical conjugation, loaded with pre-microRNA-146a and injected into mice, a wide biodistribution (except for the liver tissue) has been observed and functional pre-microRNA was successfully delivered into the cells as proven by the suppression of the production of the target protein. For 120 days, the amount of the delivered microRNA remained stable [70].

Even mammalian viruses may face a problem with limited receptor-dependent tropism. The spectrum of cell types that might be clinically relevant (e.g., for ex vivo therapy) include cell types and tissues that do not contain receptors for these viruses. CPPs, with their ability to penetrate membranes, could therefore broaden the tropism of VPs and their cellular uptake while using various ways in which to enter the cells or increasing the degree of cell attachment (Figure 3a). For example, the cells of the nervous system and immune cells have been shown to become accessible for Ad following modification of Ad with Tat and VP22 CPPs [50]. Since some cancer cells have been shown to exhibit limited permissivity due to the low expression of CAR [78], Ad provides an example of a viral vector that may benefit from such modifications with CPPs. A number of studies [44,45,47,50,52,55,56,57,58,59,60,61,62,63,65,66,68,69,70,71,73] have demonstrated that CPPs are able to greatly enhance the transduction of tumor cells derived from a broad range of tissues.

Yu et al. [58] have shown that following the transduction of several cell lines by Ad modified with Tat (Tat-Ad), the number of GFP positive cells was often close to 100% even for low-CAR expressing cells (often tumor cell lines) that exhibited very low transduction by Ad alone. High-CAR expressing cells were transduced by Tat-Ad with at least the same efficiency as that provided by Ad alone.

Park et al. [48] have reported that a branched tetramer of Tat associated with a recombinant adenovirus strongly increased the transduction of bone marrow-derived mesenchymal stem cells (BM-MSC—cells not permissive for Ad) to almost 100% compared to monomeric Tat that exhibited around 10% transduction even when used in a 1000-fold higher concentration than that of the branched Tat. Moreover, following the ex vivo transduction of the BM-MSCs described above and their implantation into rats with a calvarial defect, the formation of new bone material and bone regeneration was observed, which suggests great potential for ex vivo gene therapy with mesenchymal stem cells.

As Kühnel et al. [50] have pointed out, the approach involving the broadening of the spectrum of target cells may also be beneficial with regard to oncolytic viruses (and other tumor-targeted vectors). The specificity of oncolytic viruses depends either on receptor-mediated entry controlled by a specific molecule (receptor) overexpressed on the tumor cell surface or on conditional replication restricted only to the tumor cells (e.g., dependent on the presence of a telomerase [79]). Broadening the virus tropism to include a wide spectrum of cells would, therefore, assist in terms of hitting every tumor cell including subpopulations that do not overexpress the specific targeted molecule when a second strategy is applied. In addition, the use of CPPs may lead to a reduction in the dose of the required virus and thus minimize its possible cytotoxic effect, a concept that was demonstrated using an oncolytic Ad with the Tat peptide incorporated into a hexon protein sequence (Tat-Ad) in the following experiment [58]. Mice were subcutaneously injected with SK-N-SH neuroblastoma cells (nonpermissive for native Ad serotype 5) or CNDT2.5 neuroendocrine tumor cells in order to induce a tumor followed by the peritoneal injection of oncolytic Ad or Tat-Ad. The oncolytic Ad modified with the Tat peptide, unlike the unmodified Ad, slowed down tumor growth and improved the survival rate of the mice. Interestingly, a comparison of oncolytic activity in the neuroblastoma cells and the neuroendocrine tumor cells treated by oncolytic Ad or Tat-Ad in vitro revealed that the unmodified Tat induced viral replication to a certain extent but failed to kill the cells, whereas Tat-Ad killed the cells efficiently. Yu et al. [58] have found that CPPs are able to help overcome the attenuation of infection by oncolytic Ad in cells that surround the primarily infected site, which is usually caused by a fiber protein that is released from the infected cells [80]. Yu et al. infected the cells via Ad and Tat-Ad in the presence of a soluble fiber protein and determined that Tat-Ad transduction decreased only slightly (20%), whereas native Ad transduction decreased substantially (around 80%) compared to transduction in the absence of a fiber protein. The Tat modification of Ad was also found to reduce undesirable interaction with coagulation factor X that may re-target the virus to liver tissue in vivo [81], thus providing a further step forward in terms of the clinical application of Tat-Ad.

### 4.2. Increasing the Efficiency of VP Transduction

The penetration potential of CPPs can also be employed to increase the effectiveness of events that result in the successful delivery of a cargo by VPs into cells, which could be naturally permissive for the parent virus (Figure 3b). The underlying mechanism might again consist of stronger attachment to the plasma membrane via electrostatic interactions, the exploitation of a different receptor, and the employment of various entry pathways (Figure 4). Moreover, the higher transduction efficiency might be caused not necessarily by more frequent penetration through the plasma membrane but by more efficient release of VPs from endosomal vesicles into the cytoplasm, which facilitates the delivery of the VPs into their final destination (Figure 4c). The enhancement of transduction efficiency is desirable, e.g., when performing ex vivo therapy, which benefits from the maximum possible transduction rate. At the same time, it is important for in vivo transduction that every target cell is affected. The utilization of CPPs could minimize the dose of VPs necessary so as to provide for the required efficiency and to reduce the possible cytotoxicity of the VPs and the chance of recognition by the immune system.

Kurachi et al. [56] have determined that an adenovirus genetically modified with the Tat peptide attained a two-log order higher cell transduction value than that of unmodified Ad. Youn et al. [44] have even attained around 95% GFP positive mouse and human tumor cells when using an adenovirus carrying a GFP gene with the HP4 peptide that also enhanced the transduction level of MSCs 10-fold over recombinant Ad alone. Gratton et al. [46] have injected Ad noncovalently associated with the Pen peptide into isolated mouse skeletal muscle tissue and the carotid artery. The Ad associated with the Pen transduced the skeletal muscle cells and endothelial cells of the carotid artery to a higher extent than did Ad alone. Moreover, the suitability of these VP-CPP complexes for in vivo application was demonstrated by the injection of Ad with the gene for vascular endothelial growth factor (VEGF) in the genome in a complex with Pen peptide into a mouse with an ischemic hind limb, which led to significantly enhanced angiogenesis than did Ad alone [46].

A further example of CPP exploitation with respect to increasing transduction efficiency with VPs for possible ex vivo transduction has been demonstrated for lentiviral vectors. Lentiviral vectors pseudotyped with various retroviral glycoproteins mixed with the LAH4-L1 peptide or the LAH4-derived peptide Vectofusin-1 were seen to be able to transduce hematopoietic stem cells that have very limited permissivity for lentiviral vectors up to almost 100% depending on the pseudotype applied, which was at least as high as that attained on Retronectin-treated wells (commercially available for enhancement of lentiviral infectivity) [43,45,51]. Remarkably, in the absence of an envelope in the VP structure, LAH4-L1 did not enhance the transduction of cells [45]. It has been demonstrated that Vectofusin-1 assembles into nanofibrils that associate with the lentiviral vectors and consequently lead to an intensive sedimentation of these complexes onto the cellular membrane. In addition, Vectofusin-1 is also able to facilitate the adhesion of lentiviral VPs to the cellular membrane. Its cationic AA residues mediate fusion with the membrane by attenuating the negative charge of the membrane [51,82].

Many nanoparticles are required to deal with the problem of protein corona which form in the bloodstream around nanoparticles and prevent their functioning. Two studies [58,83] that have investigated the effect of serum on VP-CPP complexes determined a decrease in transduction in the presence of serum but to a lesser extent than that by VPs alone, which may lead to the prolongation of the circulation time in the bloodstream and thus better allow for the reaching of the cells. A greater difference in terms of neutralization by an anti-Ad serum was observed between Ad with a chemically conjugated Tat peptide and unmodified Ad [65] than between Ad with the Tat peptide incorporated in the hexon protein and the unmodified Ad [58]. In bacteriophage lambda genetically modified with Tat, the presence of the serum increased or slightly decreased the transduction of cells; the effect was dependent on the cell line [63].

Interestingly, some studies have examined whether the improvement in VP transduction using CPPs occurs via interaction with the original viral receptor or whether a new receptor plays role. Since CAR is a natural receptor for Ad serotype 5, Yoshioka et al. [65] have attempted to determine whether the transduction of an adenovirus conjugated with the Tat peptide could be blocked by the anti-CAR antibody in order to prove that the internalization of the Tat-Ad complex is dependent on CAR or whether another cell entry path exists. Unlike the unmodified Ad, Tat-Ad was still able to enter the cell in the presence of anti-CAR antibody to a similar extent, suggesting that in this case a different receptor or entry pathway was exploited.

### 4.3. Immunomodulation

CPPs are short peptides that are supposed to not elicit an immune response [84] and that have the potential to protect VPs from recognition by the immune system by hiding their immunogenic epitopes and prolonging the circulation of VPs in the body (Figure 3c left panel). This has been demonstrated for Ad conjugated with the Tat peptide that was neutralized with Ad antiserum to a lesser extent than the Ad alone [65]. Kim et al. [83] coated human Ad serotype 5 with an arginine-based polymer and determined that such a cover decreased its recognition by the host immune system since the macrophages exposed to these complexes produced a lower amount of proinflammatory interleukin 6.

Conversely, CPPs are able to transport cargoes into dendritic cells and are being studied as important inducers of both CD8 and CD4 T cell responses when linked to an antigenic cargo [85]. This is potentially important, particularly in the design of CPP-based anti-tumor vaccines, which might also benefit from the presence of multiple differing epitopes in the antigens [85]. VPs provide a platform that can be used for the construction of such multiepitopic antigens and possess per se adjuvant activity for the stimulation of innate and cell-mediated immunity [86]. Moreover, the delivery of VPs with antigens into the cytoplasm of antigen-presenting cells mediated by CPPs could result in the presentation of the incorporated epitopes on MHC class I and class II and efficiently stimulate cognate CD4+ and CD8+ T cells (Figure 3c right panel). Although various CPPs have been investigated as enhancers of cell-mediated immunity for antigenic cargoes consisting of DNA vaccines, peptides, proteins, or nanoparticles (e.g., liposomes) [85,87,88,89,90,91], the impact of CPPs on VP-based vaccines has not yet been systematically investigated. However, the proof-of-principle has been presented recently in a study concerning VPs derived from the core protein of the hepatitis B virus fused with a cell-translocation motif peptide [92]. The capsids with translocation motif loaded with antigen triggered the activation of the dendritic cells and the stimulation of the CD8+ cytotoxic T lymphocyte response as well as the specific killing of the target cells in contrast with the HBV capsid, which lacked the translocation motif. Thus, further investigation will be required of combinations of CPPs and VPs as platform technologies for the development of preventive and therapeutic vaccines.

### 4.4. Combination of Strategies and Specific Targeting

The current trend with respect to the development of nanotherapeutics involves combining several utilities in one multifunctional tool which can be achieved via the modification of the viral capsid or envelope with various peptides or other compounds via genetic modification, chemical conjugation, or noncovalent association, as discussed above. The efficient penetration of specific target cells is able to limit the side effects associated with the dissemination of a drug to healthy off-target tissue, which is vital for clinical purposes. VPs can be modified by means of targeting ligands that enable them to interact with a specific receptor on the cell or tissue of interest so as to mediate entry. Such a targeting moiety that exclusively mediates the internalization of nanoparticles by target cells might, in the broader sense, be considered a CPP [69] (Figure 3d and Figure 5a). For example, nanoparticles derived from the hepatitis B virus (HBV) have been targeted via a covalently-coupled CPP toward human carcinoma cells via interaction with an epidermal growth factor receptor [69]. Specific targeting toward tumor cells can also be directed toward the tumor microenvironment; for example, histidine-rich CPPs can be exploited for such a targeting strategy [93]. The tumor microenvironment tends to be acidic, which leads to the protonation of the histidine residues in the CPPs that preferentially penetrate the tumor cell membrane [93] (Figure 5b). Furthermore, CPPs that target specific organelles can be utilized to attain delivery into a particular cell compartment, such as the endoplasmic reticulum, Golgi apparatus, lysosome, mitochondria, or nucleus (as reviewed in [94]; Figure 5c). It has been suggested that the most efficient strategy would be to first determine the biodistribution of a particular CPP and only then focus on its utilization for treatment of a tissue-specific pathology [95].

In addition to targeting and penetrating sequences, other features can be attached to or incorporated within VPs, e.g., endosomal escape domains, activatable CPPs, i.e., CPPs that are combined with anionic peptides that prevent their penetration function and that are cleaved by proteases in the tumor microenvironment, fluorescent dyes, quantum dots, and other features that serve for diagnostics and/or therapy (as reviewed in [27,96]). Moreover, it has been shown with respect to several CPPs that the use of the cyclic instead of the linear form of these CPPs is capable of leading to more efficient cell entry than that enabled by their noncyclic variants [39].

An excellent example of the combination of several techniques has been provided by Pang et al. [73] who introduced green fluorescence VPs derived from the Qβ bacteriophage loaded with epirubicin and modified with CPP and ^68^Ga-DOTA (a substance used for positron emission tomography imaging) as promising nanotherapeutics for the convection-enhanced delivery of agents to the brain. The biodistribution of these VPs exhibited a dramatic preference for brain tumor tissue over healthy tissues (including the brain). Following the application of two doses of modified VPs, the mice with brain tumor xenografts survived for 50 days with the complete eradication of the tumor.

A further example of VPs modified by means of multiple compounds has been provided by Jamali et al. [49] who produced pseudotyped lentiviral vectors modified with targeting sequences for interaction with CD4+ and CD8+ T lymphocytes and a reporter molecule—a truncated version of the low-affinity nerve growth factor receptor on the surface of the VP. These modified VPs that were noncovalently associated with LAH4-derived peptide (Vectofusin-1) delivered the plasmid for the expression of the chimeric antigen receptor to T lymphocytes, which enabled the killing of the target tumor cells. Interestingly, although these VPs exhibited increased adhesion even to the nontarget cells, only the target cells were transduced.

## 5. Conclusions

The modification of VPs by CPPs offers a range of interesting possibilities with respect to both experimental and clinical applications. The ability of CPPs to penetrate the cell membrane and to translocate VPs can be used to broaden the range of cells that can be entered by VPs and even to enhance the transduction rate. In addition, CPPs are also able to modulate the immune system response.

The CPP-mediated enhancement of the efficiency of VP penetration also allows for the use of lower viral titers, which both reduces the consumption of materials and minimizes VP cytotoxicity, which, in turn, reduces the risks entailed in their potential clinical application.

The investigation of CPP behavior is methodologically very demanding and even following three decades of research on CPPs, a number of questions remain unanswered, e.g., what happens to the VP-CPP complex once inside the cell? Is it CPP or VP which primarily determines the trafficking pathway and the final fate of particles and their cargoes? Is the use of CPP really compatible with the targeting strategy or might CPPs actually increase the degree of entry to off-target tissues? Could CPPs improve VP-based vaccines? A better understanding of the underlying mechanism of the action of VP-CPP complexes will eventually allow their design to be tailored to particular needs.

Millions of years of evolution have shaped viruses to be ideal candidates for intracellular delivery. Although we only recently have begun to learn how to exploit the enormous potential provided by viruses, we now know that their potential is far greater than previously thought. The amazing ability of viruses to conquer cells is now being enhanced by the exceptional efficiency provided by CPPs.

## Figures and Tables

**Figure 1 materials-12-02671-f001:**
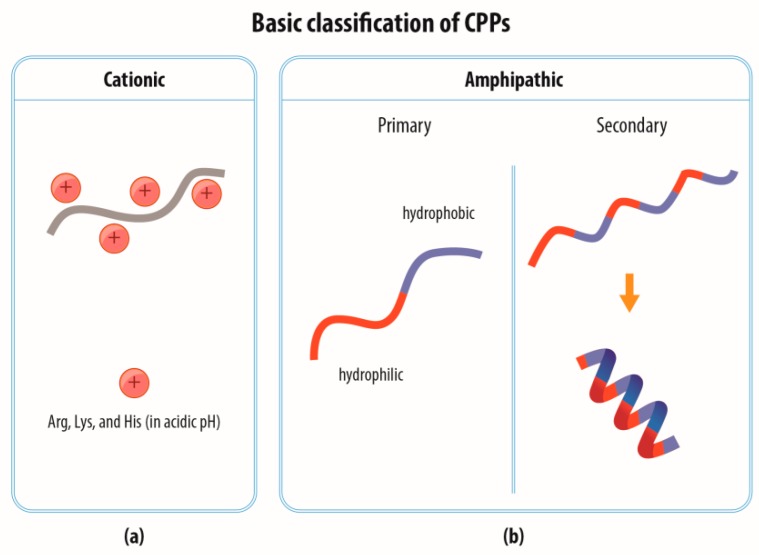
Basic classification of CPPs according to their amino acid (AA) composition. There are two main categories of CPPs: (**a**) cationic and (**b**) amphipathic. Cationic CPPs contain a high number of positively charged AA such as arginine (Arg), lysine (Lys), or histidine (His) that also becomes protonated in acidic pH. Amphipathic CPPs contain polar (hydrophilic) and non-polar (hydrophobic) regions of amino acids. In primary amphipathic CPP these two regions are distributed next to each other in the primary sequence. The secondary amphipathic CPPs form functional hydrophilic and hydrophobic regions after folding into α-helical and β-sheet-like structures. More complex organization of CPPs (e.g., combinations of stretches of cationic and amphipathic AA), however, exist (not shown).

**Figure 2 materials-12-02671-f002:**
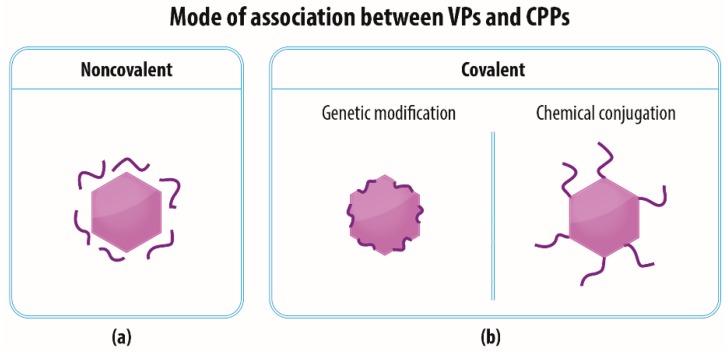
Mode of association between viral particles (VPs) and CPPs. VPs can be associated with CPPs either (**a**) noncovalently on the basis of, for example, electrostatic interactions, or (**b**) covalently via genetic modification of viral capsid proteins or chemical conjugation of CPPs to the surface of VPs.

**Figure 3 materials-12-02671-f003:**
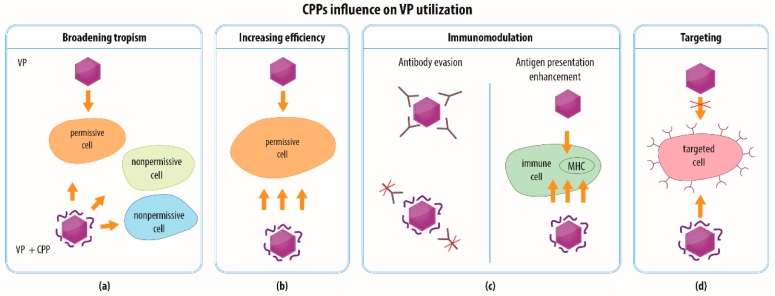
The effect of CPPs on VPs. CPPs can be utilized in order to (**a**) broaden the tropism of VPs by facilitating VP entry even to cells that are initially non-permissive for native virus, (**b**) increase the efficiency of VP transduction as specified in detail in Figure 4, (**c**) modulate the immune response to VPs either by attenuation of immune response by covering the particle surface and preventing recognition by antibodies or by inducing higher immune response via enhancement of antigen presentation on major histocompatibility complex molecules (MHC), which is especially desirable in vaccine development), and (**d**) target cells of interest as specified in detail in Figure 5, e.g., by acting as a ligand of a receptor specific for the targeted cell.

**Figure 4 materials-12-02671-f004:**
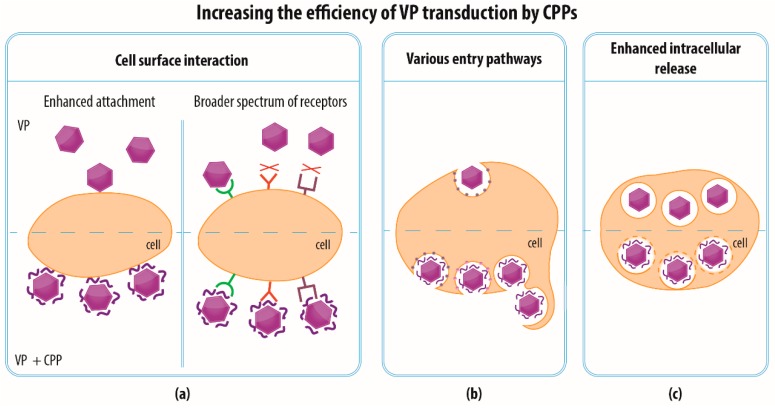
Possible mechanisms of increasing transduction efficiency of VPs by CPPs. CPPs can increase the efficiency of VP transduction during several steps. (**a**) On the cellular surface, CPPs can either strengthen the attachment of VPs to the cellular membrane or/and expand the number of cellular receptors that can be utilized by modified VPs for successful entry. (**b**) CPPs can also enable VPs to employ additional entry pathways (such as clathrin- or caveolin-mediated endocytosis, clathrin- and caveolin-independent endocytosis or macropinocytosis) which are not commonly used by the unmodified VPs. (**c**) In the cell, facilitation of the release of VPs from endocytic vesicles can prevent sequestration and degradation of VPs in lysosomes and thus significantly contribute to successful delivery of the VP cargo to its final intracellular destination.

**Figure 5 materials-12-02671-f005:**
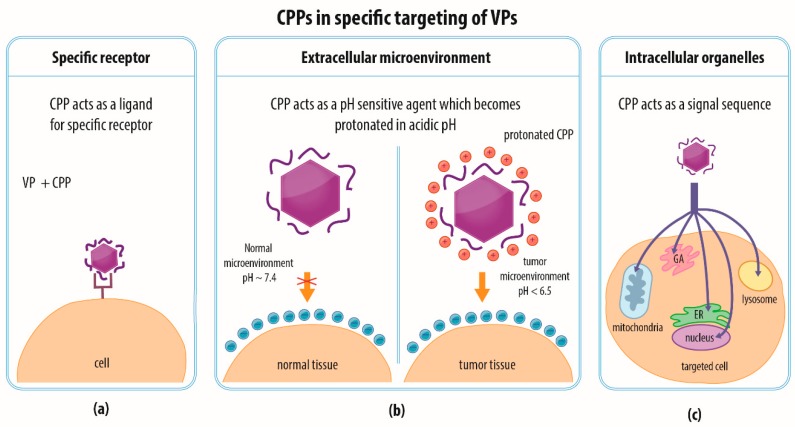
The role of CPPs in VP targeting. (**a**) CPPs can guide VPs to specific receptors and even avoid their interaction with off-target tissue to prevent their cargoes from entering those cells. (**b**) In the case of histidine-rich CPPs, their protonation in acidic pH of the tumor microenvironment enables them to penetrate into tumor cells, whereas the healthy tissues remain unaffected. (**c**) In theory, CPPs can be modified in different ways to ensure that VPs are delivered into specific organelles (Golgi apparatus (GA), endoplasmic reticulum (ER), mitochondria, lysosomes, or nucleus) as demonstrated for other cargoes [94].

**Table 1 materials-12-02671-t001:** List of cell-penetrating peptides (CPPs) mentioned in the review.

Peptide Name	Sequence	Characteristics
Tat	(Y)GKKKRRQRRR ^1^	Cationic
Penetratin (Pen) ^2^	RQIKIWFQNRRMKWKK	Cationic
Polyarginine (R_5_, R_8_ or R_9_)	RRRRR(RRR(R))	Cationic
Pep-1	KETWWETWWTEWSQPKKKRKV	Amphipathic
Proline-rich peptide (Pro)	VRLPPPVRLPPPVRLPPP	Proline-rich
Tat-HA2	CRRRQRRKKRGGDIMGEWGNEIFGAIAGFLG	Cationic fusogenic
Hph-1	YARVRRRGPRR	Cationic
HP4	RRRRPRRRTTRRRR	Cationic
LAH4	KKALLALALHHLAHLALHLALALKKA	Histidine-rich cationic amphipathic
LAH4-L1	KKALLAHALHLLALLALHLAHALKKA	Histidine-rich cationic amphipathic
Vectofusin-1	KKALLHAALAHLLALAHHLLALLKKA	Histidine-rich cationic amphipathic
Low molecular weight protamine (LMWP)	VSRRRRRRGGRRRR	Cationic

^1^ Model sequence (sequence and its length varied among studies) ^2^ Sometimes abbreviated as Antp.

**Table 2 materials-12-02671-t002:** An overview of studies focused on noncovalent association of VPs and CPPs, along with their findings.

Virus	CPP	Cargo	Cell Lines Tested	Effect	Ref.
Adeno-associated virus type 2 and 8	Pen, Tat-HA2, LAH4	Viral genome with GFP gene	HEK-293T, HepG2, NIH-3T3, BMDC, MSC, Huh7	- Increased uptake and transduction (up to 15-fold) of cell lines, primary cells, and tissues (in vivo) compared to unmodified VPs - Faster kinetics of internalization, transduction not inhibited by heparin or anti-HSPG antibody - Improved endosomal escape	[47]
Adenovirus	Tat-CAR, VP22-CAR, R_9_-CAR, Pen-CAR, Tat	*LacZ* or GFP gene or hTERT promoter in the genome	H4IIE, BNL, RT-101, T-36274, RKO, SAOS-2, SKLU-1, MCF-7, HT1080, HepG2, Huh7, HeLa, HT29, KLN205, P388D.1, RAW264.7, DC2.4, Jurkat, ISC	- Increased transduction by Tat-CAR and VP22-CAR of all permissive and non-permissive cells - Transduction inhibited by heparin (analogue of heparan sulfate)	[50]
Adenovirus	HP4, Tat, Pen, Hph-1	Viral genome with GFP or IL-12N220L gene	A375, CT26, B16F10, U-87MG, HeLa, A549, K562, C6Bu1, UCB-MSC, BM-MSC, AT-MSC, BMDC	- Increased transduction of cell lines (sometimes >95% transduction efficiency, 20-fold higher than Tat) - In mice: prolonged survival rate with tumor (80%) after injection with ex vivo transduced CT26 cells	[44]
Adenovirus	Branched oligomeric Tat, Hph-1, Pen, HP4	Viral genome with eGFP, human bone morphogenetic protein 2, or brain-derived neurotrophic factor gene	BM-MSC, UCB-MSC	- Increased internalization and transduction (>95%) of both cell lines - In rats: application of ex vivo transduced MSC led to bone reparation	[48]
Adenovirus, retrovirus	Pen, Tat	Viral genome with GFP, β-galactosidase, eNOS, or VEGF gene	COS-7, HUVEC, BAEC	- Increased transduction of cells (for Pen in HUVEC cells 10-fold higher compared to unmodified VPs) - In vitro: increased transduction of endothelial and skeletal muscle cells - In mice: increased gene delivery into the tissues led to angiogenesis in ischemic hind limb	[46]
Adenovirus, pseudotyped lentivirus	Tat from HIV-1 and HIV-2, Pen	Viral genome with GFP gene	COS-7, SKOV3.ip1, HEY, PC-3, MG-63	- Increased transduction of almost all cell lines	[52]
Pseudotyped lentiviruses and HIV-1-derived VLPs	LAH4-L1	Plasmid with eGFP gene	HCT116, HSC	- Increased transduction of HCT116 cells (up to 12-fold higher compared to unmodified VPs, reaching up to 20–35% transduction efficiency) - Transduction of nonenveloped VPs not enhanced by LAH4-L1	[45]
Pseudotyped lentiviruses	Vectofusin-1	Plasmid with GFP gene	UCB-HSC, BM-HSC, activated human T cells	- Increased transduction of all cell lines (comparable to clinically used additives in vitro, reaching up to 87% transduced UCB-HSC and 64% transduced T-cells)	[43]
Pseudotyped retroviruses	Vectofusin-1	Plasmid with eGFP gene	UCB-HSC, MPB-HSC	- Increased transduction of HSC cell lines (comparable to clinically used additives in vitro, reaching up to 80% transduced HSC cells) - Enhanced attachment and fusion - In mice: no toxicity for hematopoietic cells after injection of ex vivo transduced HSC into immunodeficient mice reconstitution of immune system	[51]
Lentiviral vectors targeted to CD4 and CD8 and pseudotyped lentiviruses	Vectofusin-1	Plasmid for expression of chimeric antigen receptor and reporter molecule: truncated version of the low-affinity nerve growth factor receptor on VP surface	Human T lymphocytes	- Increased transduction of CD4+ and CD8+ cells (2-fold reaching up to 57% of CD4+ cells and 2,7-fold reaching 87% of CD8+ cells) by targeted and CPP-modified VPs compared to unmodified VPs - Delivery of plasmid DNA enables killing of the target tumor cells - Increased adhesion even to non-target cells but transduction only of target cells	[49]

Legend: CAR, coxsackie and adenovirus receptor; eNOS, endothelial nitric oxide synthase; VEGF, vascular endothelial growth factor; A375, human melanoma cells; A549, human lung epithelial carcinoma cells (high CAR); AT-MSC, human adipose tissue-derived mesenchymal stem cells; B16F10, mouse melanoma cells; BMDC, mouse bone marrow-derived dendritic cells; BM-MSC, human bone marrow-derived mesenchymal stem cells (no CAR); BM-HSC, human hematopoietic stem cells derived from bone marrow (hCD34+); BNL, mouse hepatoma cells; C6Bu1, rat glioma cells; COS-7, African green monkey kidney fibroblast cells; CT26, mouse colon carcinoma cells (low CAR); DC2.4, mouse dendritic cells; H4IIE, rat hepatoma cells; HCT116, colon cancer cells (permissive for lentiviruses); HEK-293T, human embryonic kidney cells transformed by SV40 large T antigen; HeLa, human cervix adenocarcinoma cells (CAR positive); HepG2, liver hepatocellular carcinoma cell (almost non-permissive for AAV-2); HEY, human ovarian carcinoma cells; HSC, human hematopoietic stem cells (hCD34+); HT1080, human fibrosarcoma; HT29, colon carcinoma cells; Huh7, human hepatocellular carcinoma cells; ISC, immortalized rat Schwann cells; K562, human chronic myeloid leukemia cells; KLN205, mouse lung squamous carcinoma cells; MCF-7, human breast carcinoma cells; MG-63, osteosarcoma cells; MPB-HSC, granulocyte colony-stimulating factor-mobilized peripheral blood cells; MSC, murine primary mesenchymal stem cells; NIH-3T3, mouse fibroblast cells (almost non-permissive for AAV-2); P388D1, mouse macrophage cells; PC-3, prostate carcinoma cells; RAW264.7, mouse macrophage-like cells (CAR negative); RKO, human colon carcinoma cells; RT-101, mouse skin epidermal cells; SAOS-2, human osteosarcoma cells; SKLU-1, human lung adenocarcinoma cells; SKOV3.ip1, ovarian carcinoma cells; T-36274, mouse skin epidermal cells; U-87MG, human glioma cells; UCB-HSC, human hematopoietic stem cells derived from umbilical cord blood (hCD34+); UCB-MSC, umbilical cord blood-derived mesenchymal stem cells (no CAR).

**Table 3 materials-12-02671-t003:** An overview of studies focused on genetic modification of VPs with CPPs, along with their findings.

Virus	CPP	Place of CPP Incorporation	Cargo	Cell Lines Tested	Effect	Ref.
Baculovirus	Two longer versions of Tat	Envelope protein GP64 or capsid protein VP39	Viral genome with Luc or eGFP gene	Vero E6, U2OS, CHO-RD	- Up to 5-fold increase in transduction (including viral genome quantification) in almost all cell lines compared to unmodified virus - Enhanced co-transduction of unmodified virus by CPP-VP	[59]
Adenovirus	Tat	Fiber knob protein	Viral genome with GFP gene	RD, D65MG, U118MG, HeLa, A549	- Increased transduction of cell lines in vitro and of in vivo established tumor compared with unmodified virus - Effect decreased by soluble adenoviral receptor CAR and heparin (analogue of heparan sulfate)	[55]
Adenovirus	Tat	Fiber knob protein—HI loop or C-terminus	Viral genome with Luc gene	U937, Jurkat, CSMC, ASMC, LN444, SF295, SK HEP-1	- Increased transduction of all cell lines (sometimes two log orders higher) - In mice: similar organ distribution after systemic administration as unmodified virus	[56]
Adenovirus	Tat	Fiber knob protein—HI loop	Viral genome with eGFP gene	A549, CHO, CHO-CAR, T24, NIH-3T3, C39, HUVEC	- Increased transduction (by 30–50%) of all CAR-deficient cells - Decreased transduction of CAR-positive cells by 50% - Transduction decreased by free Tat peptide and not inhibited by soluble Ad fiber knob - Transduction was dynamin-independent	[57]
Adenovirus	Tat	Hexon protein—hypervariable region 5	Viral genome with GFP gene or complete oncolytic virus for in vivo assay	BON, CNDT2.5, SKOV-3, A549, MB49, 911, 1064SK, mel526, SK-N-SH, HUVEC	- Increased transduction (including viral genome quantification) of all cell lines compared to unmodified VPs - Cellular entry less inhibited by soluble fiber - Decreased factor-X-mediated binding to SKOV-3 cells compared to unmodified VPs - Slightly lower neutralization by anti-Ad plasma than the unmodified VPs - In mice: reduced growth of neuroendocrine and neuroblastoma tumor and prolonged survival	[58]
MS2 bacteriophage-derived VLPs	Tat	Tat incorporated via a linker at the N-terminus of coat protein	Pre-microRNA-122	Hep3B, Huh7, HeLa, HepG2, Huh7	- Increased and dose-dependent delivery of microRNA122 in all cell lines leading to about 20% decreased migration, about 30% decreased invasion, and induction of apoptosis of cells - In mice: transduction of microRNA122 leads to inhibition of hepatocellular carcinoma growth	[62]
Phage lambda	Longer version of Tat	D protein—N-terminus	Viral genome with eGFP or Luc gene	COS-1, VA13/2RA, HEK-293, NIH-3T3, HeLa, A431	- Increased transduction of mammalian cell lines (one to three log orders higher Luc activity) compared to unmodified VPs - Strong GFP signal after transduction in vivo compared to unmodified VPs observed on tissue sections - Increased or slightly decreased transduction of cells in the presence of serum, depending on the cell line - Transduction inhibited by anti-Tat Ab, heparin, and dextran sulfate - Transduction (caveolae-mediated) moderately inhibited by nystatin and filipin - Transduction occurs even at 4 °C	[63]
Recombinant bacteriophage PP7-derived VLPs	Tat	Coat protein	Pre-microRNA-23b	SK-HEP-1, COS-7	- Increased penetration of cells compared to unmodified VPs (microscopic evaluation) - Increased delivery of pre-microRNA-23b leading to reduction of migration of hepatoma cells	[60]
Recombinant bacteriophage PP7-derived VLPs	LMWP	Coat protein	mRNA encoding GFP protein	RM-1	- Increased penetration of cells compared to unmodified VPs (microscopic evaluation) - Successful delivery of GFP mRNA and expression of GFP gene	[61]

Legend: Luc, luciferase; anti-Ad, antibody against adenovirus; Ab, antibody; 911, human embryonic retinoblasts (HER) transformed by a plasmid containing base pairs 79-5789 of the Ad5 genome; 1064SK, cell derived from human foreskin (low CAR); A431, human squamous carcinoma cells; ASMC, aortic smooth muscle cells; BON, human carcinoid cells (high CAR); C39, human fibroblast cells; CHO, Chinese hamster ovary cells (low CAR); CHO-CAR, Chinese hamster ovary cells (high CAR); CNDT2.5, human midgut carcinoid cells (low CAR); COS-1, African green monkey kidney fibroblast cells; CSMC, coronary smooth muscle cells; D65MG, human glioma cells; Hep3B, human hepatocellular carcinoma cells; HUVEC, human umbilical vein endothelial cells (moderate CAR); LN444, glioblastoma multiforme cells (CAR negative); MB49, urothelial carcinoma cells (low CAR); mel526, melanoma cells (moderate CAR); RD, embryonic rhabdomyosarcoma cells; RM-1, mouse prostate cancer cells; SF295, glioblastoma multiforme cells (CAR negative); SK HEP-1, hepatoma cells (CAR+); SK-N-SH, human neuroblastoma cells (low CAR); SKOV-3, ovarian cancer cells; T24, human prostate cancer cells; U118MG, human glioma cells; U2OS, human bone osteosarcoma epithelial cells; U937, histiocytic lymphoma cells; VA13/2RA, human fibroblasts; Vero E6, African green monkey kidney epithelial cells.

**Table 4 materials-12-02671-t004:** An overview of studies focused on chemical conjugation of CPPs onto VPs, along with their findings.

Virus	CPP	Cargo	Cell Lines Tested	Effect	Ref.
Adenovirus	Tat, R_8_	Viral genome with GFP or Luc gene	A549, HeLa, U937, B16BL6, CT26, RAW264.7, EL4, LN444, LNZ308, SF295	- Increased transduction of CAR-negative and blood cell lines (one to three log orders higher) - Transduction not increased in CAR-positive cell lines - Lower neutralization by anti-Ad and anti-CAR antibodies compared to unmodified Ad - Transduction (macropinocytosis) inhibited by amiloride and by heparin	[65]
Adenovirus	Tat, R_8_, Pro	Viral genome with Luc gene	A549, CT26, B16BL6	- Increased transduction of CAR-negative cell lines (one to two log orders higher) compared to unmodified VPs - Transduction (macropinocytosis) by Tat-Ad and R8-Ad decreased by amiloride - Transduction by Tat-Ad decreased by heparin (analogue of heparan sulfate) - Transduction by R8-Ad decreased by chondroitin sulfate B	[66]
Adenovirus	Pen, Tat, R_9_, Pep1	Viral genome with *LacZ* gene	NIH-3T3	- Increased transduction (up to 80-fold) compared to unmodified VPs - Electrostatic and/or hydrophobic interactions with cells	[67]
Cowpea mosaic virus-derived VPs	R_5_	No	HeLa	- Increased penetration into cells (up to eight times higher) compared to unmodified VPs - Energy-dependent internalization - Decreased retention in endolysosomal vesicles (Lamp-1 colocalization)	[68]
Hepatitis B VPs	NRPDSAQFWLHH	No	A431	- Increased penetration into cells compared to unmodified VPs (microscopic evaluation)	[69]
MS2 bacteriophage-derived VPs	Tat	Pre-microRNA-146a	HeLa, HepG2, Huh-7, PBMC	- Increased and dose-dependent delivery of microRNA122 in all cell lines (up to about 15-fold) - Effect stable for 120 h - Delivery of miRNA increased in plasma, lung, spleen, and kidney, and almost not in liver - In vitro: transduction leads to suppression of the activity of reporter gene and of interleukin-1 receptor-associated kinase 1 - In mice: transduction leads to suppression of the activity of interleukin-1 receptor-associated kinase 1	[70]
MS2 bacteriophage-derived VPs	Tat	Antisense RNA against hepatitis C virus regulatory regions	Huh-7	- Increased penetration of Tat-modified VPs into cells compared to unmodified VPs (microscopic evaluation) - Decreased and dose-dependent expression of respective control gene	[71]
P22 bacteriophage-derived VPs	Tat	Ziconotide peptide	RBMVEC	- Translocation through the rat and human mimics of the blood brain barrier in vitro and in vivo - Colocalization with LysoTracker - Decreased penetration (clathrin-mediated endocytosis) by hypertonic solution - Colocalization with recycling endosomes (Rab11 protein marker)	[72]
Qβ bacteriophage-derived targeted VPs	KYGRRRQRRKKRG	Epirubicin, GFP	GBM U87-MG	- Increased transduction of cells (two-fold higher compared to unmodified VPs) - In mice: preference for tumor tissue, after two doses of modified VPs complete tumor eradication and survival with brain tumor xenograft	[73]
Turnip yellow mosaic virus	Tat, R_8_, Pep-1, Pen	Fluorescein dye conjugated to the interior of the capsid	BHK	- Increased transduction of cells compared to unmodified VPs and lipofection, apart from Pep-1 - Delivery of fluorescein dye into cells	[74]

Legend: Ad, adenovirus; B16BL6, mouse melanoma cells (no CAR); BHK, baby hamster kidney cells; EL4, mouse lymphoblast cells; GBM U87-MG, glioma cells; LNZ308, glioblastoma multiforme cells (CAR negative); PBMC, peripheral blood mononuclear cells; RAW264.7, mouse macrophage-like cell line (CAR negative); RBMVEC, rat brain microvascular endothelial cells.

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
