# Peer review of "The Utilization of Cell-Penetrating Peptides in the Intracellular Delivery of Viral Nanoparticles"

_materials, 2019, doi:10.3390/ma12172671_

Round 1

Reviewer 1 Report

Although briefly (this rather a mini-review), current article outlines broad spectrum of conditions and species illustrating use of large variety of recently developed applications and approaches based on utilization of limited range of cell-penetrating peptides (CPP) in combination with viral particles (VP). Authors critically analyzed existing and described approaches widely used in the development of new therapeutic agents and materials. The review presents data on the classification and internalization of CPP on the types of VPS / CPPS associations and the conditions of CPP-functionalized VPs. This review objectively reflects both the description of a number of existing problems and critically analyses potential ways and strategies for solving them. The language of the review easy to understand and individual sections/chapters are logically and coherently interrelated. In summary, current review based on present-day literature will acclaim high demand and certainly will be useful to the field.

Minor comment and recommendation to update References list and include Ülo Langel’s recent publication - https://doi.org/10.1007/978-981-13-8747-0

I recommend this review for “Materials”. 

Author Response

We thank the Reviewer 1 for her/his careful reading of the manuscript and for the pleasant commentary.  We updated the Reference list and included recommended publication that appears on the line 88 in the main text and as reference number 11 in the Reference list.

Reviewer 2 Report

The review is overall comprehensive. However, a few more figures (at least three more) as illustration will be helpful to present the review better. 

Author Response

We thank the Reviewer 2 for her/his careful reading of the manuscript and her/his suggestion. We prepared three more figures to illustrate (i) Basic classification of CPPs according to their amino acid (AA) composition (Figure 1), (ii) Possible mechanisms of increasing transduction efficiency of VPs by CPPs (Figure 4) and (iii) The role of CPPs in VP targeting to specific cells (Figure 5). We agree that these illustrations helped to present the review better.

Reviewer 3 Report

Authors have proposed a Review on viral nanoparticles conjugated to cell penetrating peptides.

The manuscript is timely, well-organized, and the topic interesting. My only concern is about its presentation. Indeed, in some point the manuscript is hard to follow, and some concepts may be lost from Readers. Overall, I suggest its acceptance after the Authors perform a moderate re-styling of the presentation. 

Author Response

We thank the Reviewer 3 for her/his careful reading of the manuscript a her/his suggestions of re-styling the presentation. Since the Reviewer did not mention the specific parts of manuscript we decided to improve the presentation by including several illustration (as also suggested by the Reviewer 2) supplemented with Legends that could help Readers to follow some concepts of our review. Specifically, we added figures to illustrate (i) Basic classification of CPPs according to their amino acid (AA) composition (Figure 1), (ii) Possible mechanisms of increasing transduction efficiency of VPs by CPPs (Figure 4) and (iii) The role of CPPs in VP targeting to specific cells (Figure 5). We hope that these illustrations will help to present the review better.

Reviewer 4 Report

The review reflects the current state of viral particules in the use of therapies. I think it reflects different approximations of his employment and the modifications to which he was subjected

Author Response

We would like to thank the Reviewer 4 for her/his careful reading of the manuscript and for the appreciative commentary. We also thank Reviewer for pointing to the error in our manuscript that has been corrected - the name of the first author in citation on line 267 (now on line 280) was deleted and only the reference number was left.

Round 2

Reviewer 2 Report

All set